# VIDEOTREE: ADAPTIVE TREE-BASED VIDEO REPRESENTATION FOR LLM REASONING ON LONG VIDEOS

## ABSTRACT

Long-form video understanding has been a challenging task due to the high redundancy in video data and the abundance of query-irrelevant information. To tackle this challenge, we propose VIDEOTREE, a training-free framework which builds a query-adaptive and hierarchical video representation for LLM reasoning over long-form videos. First, VIDEOTREE extracts query-relevant information from the input video through an iterative process, progressively refining the selection of keyframes based on their relevance to the query. Furthermore, VIDEOTREE leverages the inherent hierarchical structure of long video data, which is often overlooked by existing LLM-based methods. Specifically, we incorporate multi-granularity information into a tree-based representation, allowing VIDEOTREE to extract query-relevant details from long videos in a coarse-to-fine manner. This enables the model to effectively handle a wide range of video queries with varying levels of detail. Finally, VIDEOTREE aggregates the hierarchical query-relevant information within the tree structure and feeds it into an LLM reasoning model to answer the query. Our experiments show that our training-free method improves both reasoning accuracy *and* efficiency compared to existing methods. Specifically, VIDEOTREE outperforms the existing training-free approaches on the popular EgoSchema and NExT-QA benchmarks with less inference time, achieving $61.1\%$ and $75.6\%$ accuracy on the test set without additional video-specific training. Moreover, on the long split of Video-MME benchmark (average 44 minutes), the training-free VIDEOTREE framework achieves better performance than the strong proprietary GPT-4V model and other MLLMs that were extensively trained on video data. Our code is provided in the supplementary and will be made public.

## 1 INTRODUCTION

With the surge in accessible long video content and the growing importance of applications such as long-form human behavior analysis and movie analysis, developing models capable of reasoning over and answering questions about long-form videos has become increasingly crucial. Recently, several approaches (Zhang et al., 2023a; Wang et al., 2024g; Kahatapitiya et al., 2024) have emerged that leverage the long-sequence reasoning capabilities of Large Language Models (LLMs) to tackle the challenge in long-form video understanding in a training-free manner. . Typically, these approaches leverage vision-language models (VLM) to caption densely sampled frames, thus representing the video in text format. This text representation is then subsequently fed into an LLM, which reasons over the video and responds to the provided query. Although this strategy has demonstrated great potentials on long-form video understanding benchmarks, it still faces two major limitations:

**1) Informational Overload:** Long videos inherently contain high levels of information redundancy, and current approaches (Zhang et al., 2023a; Chung & Yu, 2023) lack a principled method to effectively address this challenge. A deluge of redundant and irrelevant information can overwhelm the LLM, leading to mistakes in long-form video reasoning and reduced efficiency.

**2) Inability to Capture the Coarse-to-Fine Video Structure:** Existing approaches (Zhang et al., 2023a; Wang et al., 2024c) often simplify video content into a list of captions without any structure, failing to account for the hierarchical nature of video information. Especially in long videos, some video regions are information-dense – requiring fine-grained temporal understanding – while others are irrelevant to the query, or information-sparse. Because of this, existing approaches not only suffer

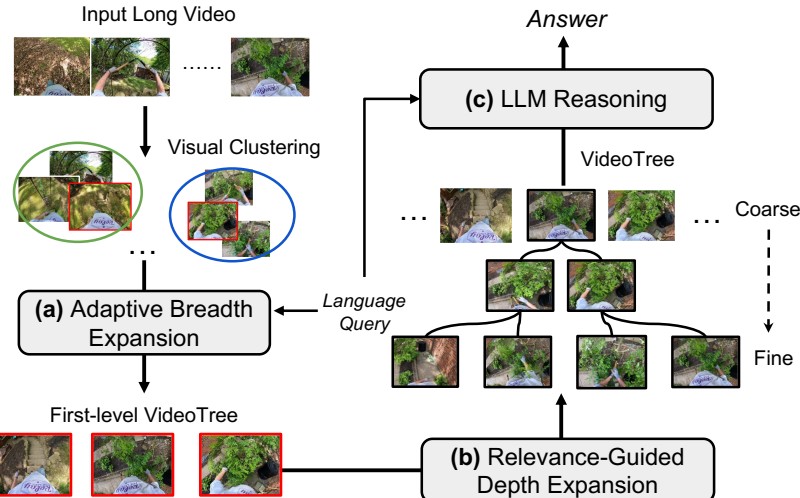

Figure 1: Overview of VIDEOTREE for LLM reasoning on long videos. Given the long video input, we first apply adaptive breadth expansion to identify the first-level keyframes for VIDEOTREE. Next, we use relevance-guided depth expansion to explore the inherent hierarchical structure of the video, forming a tree-based representation. Finally, the coarse-to-fine information extracted by VIDEOTREE is fed into the LLM reasoner to answer the query.

from the information overload problem mentioned above, but also omit key detailed information from the captions.

These limitations underscore the pressing need for a new long-form video understanding method. To this end, we introduce **VIDEOTREE**, a training-free framework for long-form video understanding. VIDEOTREE dynamically extracts query-relevant keyframes from the video input in a coarse-to-fine manner and organizes them within a tree structure, with child nodes representing more fine-grained information. VIDEOTREE is *adaptive*, meaning that our method allocates more frames to relevant video regions and fewer frames to irrelevant ones based on the given query. VIDEOTREE is also *hierarchical*. Unlike existing approaches (Zhang et al., 2023a; Wang et al., 2024c), which treat video as a list of frames, our method explores the inherent structure within the video data (e.g., events, scenes) to extract fine-grained information relevant to the video query.

VIDEOTREE relies on three crucial steps: **adaptive breadth expansion** (Fig. 1a), **relevance-guided depth expansion** (Fig. 1b), and **LLM-based reasoning** (Fig. 1c). To address redundancy in long videos, VIDEOTREE first leverages an adaptive breadth expansion module to extract query-relevant information, forming the initial level of representation. We utilize an iterative process of visual clustering, keyframe captioning, and relevance scoring until sufficient query-relevant information is gathered. Compared to existing approaches (Kahatapitiya et al., 2024; Zhang et al., 2023a) that rely on dense frame captions, VIDEOTREE selects only sparse keyframes for captioning, which significantly improves inference efficiency and helps avoid irrelevant information that could interfere with accurate video reasoning. To capture more fine-grained information, we introduce a relevance-guided depth expansion step that adds finer, query-specific details in a hierarchical structure, forming a tree-based representation. Finally, we generate video descriptions from the structured representation using a captioner and provide them, along with the query, to the LLM for long video reasoning.

We demonstrate the effectiveness and efficiency of VIDEOTREE by evaluating it on two mainstream long video question answering (LVQA) datasets, EgoSchema (Mangalam et al., 2024)and NExT-QA (Xiao et al., 2021). Compared existing training-free approaches, VIDEOTREE achieves 2.1% and 4.3% improvements on EgoSchema(subset) and NExT-QA validation set with less inference time or LLM calls. To further validate VIDEOTREE effectiveness on very long videos, we test our method on the long split of the recent Video-MME benchmark (Fu et al., 2024a) and VIDEOTREE achieves better performance than the strong proprietary GPT-4V model. Our ablation studies show that VIDEOTREE outperforms the the same category methods (VideoAgent (Wang et al., 2024c) and LLoVi (Zhang et al., 2023a)) under all number of captions and observes better efficiency-effectiveness trade-off. We further provide addition results on open-source LLM, where VIDEOTREE shows strong generalization

ability across different language backbone models and achieves $4.8\%$ improvements against the LangRepo approach (Kahatapitiya et al., 2024).

## 2  RELATED WORK

**Structural Video Representation.** Video understanding (Lin et al., 2019; Wang et al., 2023b; Li et al., 2024b; Lin et al., 2023d; Xu et al., 2023; Lin et al., 2023a; Ma et al., 2023; Wang et al., 2023d;a; Wu et al., 2022; Ren et al., 2024; Song et al., 2024; Liu et al., 2022) has shown impressive advancement in both views of comprehension and efficiency. Recently, several video-language methods (Ashutosh et al., 2023; Li et al., 2020; Islam et al., 2024; Wang et al., 2023e; Zhang et al., 2018; Zala et al., 2023; Qing et al., 2022; Sanders et al., 2024; Yang et al., 2024; Xiao et al., 2022a; Lu et al., 2022) have further introduced a structured understanding of video frames to allow compact and efficient recognition of scene contexts. For example, HierVL (Ashutosh et al., 2023) proposes a *bottom-up* hierarchical video-language embedding that capture video representations across short and long time periods. VideoReCap (Islam et al., 2024) introduces a progressive video captioning approach that generates short clip-level captions and summarizes them into longer segments. These methods process long videos by progressively building high-level knowledge from local temporal information, i.e. in a bottom-up fashion that first captures all low-level details and then aggregates. This results in significant computational and time overhead. In contrast, VIDEOTREE employs a top-down approach with dynamic depth, enabling efficient and effective long video understanding by dynamically extracting query-relevant keyframes in a coarse-to-fine manner for LLM reasoning .

**Video Understanding with LLMs.** Inspired by the powerful reasoning capabilities of LLMs, recent works have explored using LLMs to address complex video-related tasks. Since LLMs primarily process text, various methods (Munasinghe et al., 2023; Lin et al., 2023b; Korbar et al., 2024; Weng et al., 2024; Maaz et al., 2024; Zhang et al., 2023b; Tan et al., 2024; Chen et al., 2023a; Li et al., 2024a; Jin et al., 2024; He et al., 2024; Li et al., 2024d; Wang et al., 2024h; Li et al., 2024c; Yu et al., 2024b) have been developed to efficiently train multimodal projectors to connect the visual encoder and LLMs or leverage caption-centric information. Past works (Wang et al., 2022a; Kahatapitiya et al., 2024; Fan et al., 2024; Wang et al., 2024c; Surís et al., 2023; Choudhury et al., 2023; Wang et al., 2023c; Ko et al., 2023; Wang et al., 2024g) has investigated training-free combinations of captioners and LLMs for video understanding. Specifically, LLoVi (Zhang et al., 2023a) proposes a simple language-guided video understanding method. First, it extracts short-term video descriptions with a captioning model, and then an LLM summarizes these dense captions and responds to the given prompt. VideoAgent (Wang et al., 2024c) introduces a multi-round frame search strategy using an LLM agent. Different from VideoAgent's non-hierarchical keyframe searching, we propose to extract the key information long videos in an adaptive and coarse-to-fine manner, which improves the efficiency and generalize better to more frames and getting better performance.

## 3  VIDEOTREE: ADAPTIVE TREE-BASED REPRESENTATION FOR LONG VIDEO-LANGUAGE UNDERSTANDING WITH LLMS

We present VIDEOTREE, a framework for constructing a query-adaptive, hierarchical video representation for efficient LLM reasoning over long videos. As illustrated in Fig. 2, the VIDEOTREE framework consists of three main steps: adaptive breadth expansion, relevance-guided depth expansion, and LLM video reasoning. Given the highly redundant nature of long videos, VIDEOTREE first leverages an adaptive breadth expansion module to extract query-relevant information from the video, forming the initial level of representation (Sec. 3.1). To capture finer-grained details, we propose a relevance-guided depth expansion module that progressively adds finer-grained, query-specific details to in a hierarchical manner, forming a tree-based representation (Sec. 3.2). Finally, we extract the video description from the constructed representation using a captioner and feed it, along with the query, into the LLM for long video reasoning (Sec. 3.3).

### 3.1  ADAPTIVE BREADTH EXPANSION

Video data is often highly redundant, and long videos can contain substantial amounts of irrelevant information relative to the given video query. Addressing this redundancy and filtering out irrelevant

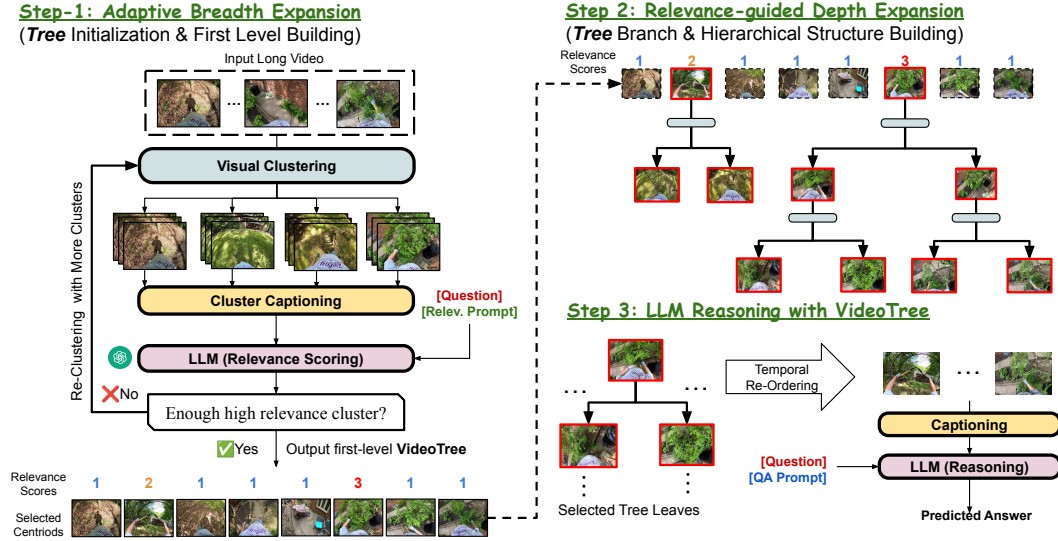

Figure 2: A detailed view of VIDEOTREE. To construct the tree structure, we begin with *Adaptive Breadth Expansion* (Step 1), which dynamically extracts query-relevant key information by considering both video and question inputs. Then, starting from the highly relevant root nodes, we explore deeper into the tree branches with *Relevance-guided Depth Expansion* (Step 2), re-clustering at each level to capture finer visual information. Finally, we gather the selected nodes (keyframes), caption them, and arrange them in temporal order for *LLM reasoning* (Step 3).

content is crucial for efficient and effective long video understanding. Existing approaches (Yu et al., 2024a; Wang et al., 2024d) select a fixed number of keyframes as the key information. However, as discussed in Sec. 1, this fixed keyframe selection is sub-optimal for a general long video-language understanding framework, since the information density varies across videos—some contain numerous scene changes, while others remain largely static. To address this, we propose an adaptive breadth expansion module that constructs the first level of the tree representation by dynamically identifying keyframes that are relevant to the given query. Specifically, as shown in the left of Fig. 2 (Step 1), given the video and a query about it, we build the first level of the tree by iterating three operations: **visual clustering**, **cluster captioning,** and **relevance scoring**. These operations first group similar frames together, then generate captions for each cluster, and use the LLM to determine how relevant each cluster is to the query. VIDEOTREE iterate these operations until getting enough query-relevant information from long videos in an *adaptive* manner. In the following paragraphs, we motivate and introduce each operation in detail.

**Visual Clustering.** To reduce the redundancy, we first propose a visual clustering operation that groups the video frames based on semantic similarity, allowing the model to focus on representative frames from each cluster while discarding repetitive or irrelevant content. Specifically, given a video sequence $V = (F_1, F_2..., F_n)$, where $F_i$ is the frame at the time step $i$ and $n$ is the length of the video, we extract visual features for each frame with the pre-trained visual encoder (Sun et al., 2024) $E$, such that $f_i = E(F_i)$, where $f_i \in \mathbb{R}^d$ is the visual features extracted by the frame $F_i$. These features serve as a compact representation of each frame's visual content, capturing diverse semantics of each frame, such as scenes and objects. Then we use K-Means clustering (MacQueen et al., 1967), to group frame features into $k$ distinct clusters, which we denote as:

$$(C_1, C_2, ...C_k), (c_1, c_2, ...c_k) = \text{K-Means}((f_1, f_2, ..., f_n), k) \quad (1)$$

where, $C_i$ is the $i$th cluster that groups multiple frames, $c_i$ is the centroid vector for the $i$th cluster and $k$ is the number of clusters. This clustering process reduces the redundancy within the video by converting the input from $n$ frames into $k$ clusters of similar frames (where $n \gg k$), effectively summarizing the video into $k$ keyframes (cluster center frame) that capture the essential semantics.

**Cluster Captioning.** To better extract the key semantics from each cluster, we leverage a captioner to convert the keyframe from each cluster to a textual description. Specifically, for the cluster $C_i$, we

find the keyframe $F_i$ that is closest to the centroid vector $c_i$ and consider it as the keyframe of the $i$th cluster. We then feed the extracted keyframe into the VLM-based captioner $Cap(\cdot)$ (Zhao et al., 2023; Liu et al., 2024) and obtain a text caption $t_i = Cap(F_i)$ for each cluster. These text captions serve as detailed descriptions of the key semantics from the corresponding clusters.

**Relevance Scoring.** To encourage the model to extract query-relevant information, after obtaining the cluster captions $t$, we leverage the reasoning capability of the LLM to assess whether the extracted information are sufficient for answering the given query. To this end, we first feed all cluster captions $\{t_i \ \forall i \in [1, \ldots, k]\}$ from the last operation and the video query $Q$ into the LLM and output a set of relevance scores $\{r_i \ \forall i \in [1, \ldots, k]\}$ for each cluster, where $r_i$ is the relevance of the $i_{th}$ cluster. Specifically, to obtain each $r_i$, we prompt the LLM with the captions and the query, asking it to assign a relevance score to each caption, with three levels: 1 (not relevant), 2 (somewhat relevant), and 3 (highly relevant). See Tab. 14 for all detailed prompts.

Then, we adaptively extract the query-relevant information within the video by iterating the clustering, captioning, and relevance scoring operation. Specifically, given the list of relevance scores for each cluster, we set a threshold of the number of highly relevant clusters $rele\_num\_thresh$ to decide the stop of the adaptive process. We also set a maximum value for the number of clusters ($max\_breadth$) to avoid infinite loops. If the number of highly relevant clusters is below the requirement, that indicates the information extracted from the current cluster assignment is insufficient for the LLM to answer the video query. In that case, we increase the number of clusters $k$ by double the original number and repeat the clustering, captioning, and relevance scoring operations. If the number of high-relevance clusters meets the threshold $rele\_num\_thresh$ or the number of clusters reaches $max\_breadth$, we append the extracted clusters with their keyframes to the first layer of the tree and continue to the next step (see more details in lines 2-11 in Algorithm 1).

### 3.2 RELEVANCE-GUIDED DEPTH EXPANSION

After obtaining the first-level clusters and their keyframes, VIDEOTREE captures high-level query-relevant information from the video input. However, some video regions are information-dense and critical for answering the query, requiring a more detailed selection of keyframes.

Existing approaches, such as SeViLA (Yu et al., 2024a) and VideoAgent (Wang et al., 2024c), typically treat the selected frames as an unstructured list, overlooking the potential internal structure within the video data. To address this, as shown in Step 2 of Fig. 2, we construct a hierarchical video representation on top of the clusters from the previous breadth expansion step, allowing us to efficiently extract query-relevant details by leveraging the semantic relationships within the video data. Specifically, we expand the depth of the tree by sub-clustering the clusters with higher relevance scores from the first step. The intuition is that for high-relevance clusters, the LLM requires more detailed, granular information, while for low-relevance clusters, more information could actually lead to irrelevant details and could overwhelm the LLM, leading to incorrect reasoning.

To build the hierarchical structure, we use the relevance of a top-level cluster to determine how many levels of more granular information will be extracted from it. Since the relevance score $r$ falls into one of three levels, we handle each first-level cluster differently based on its assigned relevance level. For "somewhat relevant" clusters, we re-cluster the first-level cluster into $w$ sub-clusters, where $w$ represents the tree's branch width, ensuring that more keyframes are allocated to these moderately relevant clusters. For "highly relevant" clusters, we re-cluster into a two-level tree with a branch width of $w$ using hierarchical clustering. This coarse-to-fine exploration strategy allows for the detailed extraction of relevant information, supporting comprehensive video analysis for complex queries. We repeat this process for all first-level clusters and build the hierarchical structure of VIDEOTREE (lines 12-15 in Algorithm 1). After the breadth and depth expansion steps, we obtain the tree-based video representation for LLM reasoning over the long video.

### 3.3 LLM VIDEO REASONING

Finally, in order to use the LLM's ability on video reasoning, we need to present the LLM with a text-based video description. To this end, we traverse the nodes of the tree starting at the roots and expanding to the leaves, extracting keyframes from the tree's clusters at all levels and passing them into the captioner to obtain keyframe captions. We then sort these keyframe captions in temporal

order and concatenate them into a textual description of the video. Finally, we pass this description and the input query to the LLM and output the final answer (see line 16-18 in Algorithm 1). Our full prompt is in Tab. 15.

## 4  EXPERIMENTAL SETUP

**Tasks & Datasets.** We test VIDEOTREE on three diverse long-form video question-answering benchmarks: (1) **EgoSchema** (Mangalam et al., 2024), a long-range video question-answering benchmark consisting of 5K multiple choice question-answer pairs spanning 250 hours of video and covering a wide range of human activities. Our ablation studies are conducted on the official validation set of EgoSchema which contains 500 questions (referred to as the EgoSchema Subset). The videos are 180 seconds long on average. (2) **NExT-QA** (Xiao et al., 2021), a video question-answering benchmark for causal and temporal reasoning. It contains 5440 videos with an average length of 44s and approximately 52K questions. NExT-QA contains 3 different question types: Temporal (Tem.), Causal (Cau.), and Descriptive (Des.). (3) **Video-MME** (Fu et al., 2024a) is a recent-proposed comprehensive evaluation benchmark for video analysis. We test VIDEOTREE on the "long-term videos" split of the dataset (long split), whose average video length is 44 minutes, ranging from 30-60 minutes.

**Implementation Details.** We adopt GPT-4[1] (OpenAI, 2023b) as our LLM for all the main results. We also provide the results with open-source LLM (Sec. 5.2) and other proprietary LLMs (Sec. C). Following VideoAgent (Wang et al., 2024c), we leverage EVA-CLIP-8B (Sun et al., 2024) as our visual encoder and also provide experimental analysis with smaller visual encoder in Sec. 5.2. Following VideoAgent (Wang et al., 2024c), we leverage CogAgent (Hong et al., 2023) as the captioner for NExT-QA benchmark and use LaViLa (Zhao et al., 2023) as our captioner for the EgoSchema benchmark due to its ego-centric video pretraining (we also show results in Tab. 12 using a unified captioner (LLaVA1.6-7B (Liu et al., 2024)) for all benchmarks). For Video-MME, we directly use the default unified LLaVA1.6-7B captioner. We preprocess videos by simply sampling the original frames at 1FPS for EgoSchema and NExT-QA benchmark and 0.125 FPS for Video-MME. The best-performing average number of captions for EgoSchema subset, Next-QA and Video-MME is 62.4, 12.6 and 128, respectively. We ablate the hyper-parameter choices in detail in Sec. C.

**Evaluation Metrics.** We evaluate VIDEOTREE on all datasets under the multiple-choice QA setting. We utilize standard accuracy metrics for all experiments.

## 5  EXPERIMENTS

### 5.1  COMPARISON WITH EXISTING APPROACHES

**Comparison with training-free methods.** Tab. 1 shows a comparison of the existing training-free works and VIDEOTREE on EgoSchema and NExT-QA benchmarks. We compare our methods with three types of systems: those using all open-source LLMs (Ranasinghe et al., 2024; Kahatapitiya et al., 2024; Shang et al., 2024), those with proprietary MLLMs (Kim et al., 2024; Park et al., 2024), and the most similar class to ours, which consists of methods with open-source captioners and proprietary LLMs (Choudhury et al., 2023; Zhang et al., 2023a; Min et al., 2024; Wang et al., 2023c; Fan et al., 2024; Wang et al., 2024c;g). Specifically, compared with the methods that leverage the same VLM (captioner) and LLM (Zhang et al., 2023a; Wang et al., 2024c;g), VIDEOTREE significantly outperforms these methods on both EgoSchema and NExT-QA benchmarks. Comparing with VideoAgent (Fan et al., 2024) which also uses video-specific models (Video-LLaVA (Lin et al., 2023a), ViCLIP from InternVid (Wang et al., 2024e)) which were trained on extensive video data, VIDEOTREE still performs better on EgoSchema. Moreover, comparing with the methods that utilize strong multimodal LLMs, VIDEOTREE significantly outperforms IG-VLM (Kim et al., 2024) (based on GPT-4V(OpenAI, 2023a)) on both EgoSchema and NExT-QA benchmarks and obtains comparable results on the EgoSchema full test set compared to the recent LVNet (Park et al., 2024) (which uses the more powerful GPT-4o for both captioner and LLM) while outperforming LVNet on NExT-QA benchmarks. Additionally, we observe a significant gap between VIDEOTREE and the open-source LLM-based approaches, highlighting the need of strong LLM reasoning module in our method. For

---

[1]version 1106

Table 1: Comparison with other training-free methods on EgoSchema and NExT-QA. We compare VIDEOTREE framework with three different types of existing methods (all open-source, all proprietary, mixed) and show the effectiveness of VIDEOTREE against all three types.

| Model | (M)LLM | EgoSchema | | NExT-QA | | | |
|---|---|---|---|---|---|---|---|
| | | Sub. | Full | Tem. | Cau. | Des. | Avg. |
| *Based on Open-source Captioners and LLMs* | | | | | | | |
| MVU (Ranasinghe et al., 2024) | Mistral-13B | 60.3 | 37.6 | 55.4 | 48.1 | 64.1 | 55.2 |
| LangRepo (Kahatapitiya et al., 2024) | Mixtral-8×7B | 66.2[1] | 41.2 | 51.4 | 64.4 | 69.1 | 60.9 |
| Video-LLaVA+INTP (Shang et al., 2024) | Vicuna-7B v1.5 | - | 38.6 | 58.6 | 61.9 | 72.2 | 62.7 |
| *Based on Proprietary MLLMs* | | | | | | | |
| IG-VLM (Kim et al., 2024) | GPT-4V | 59.8 | - | 63.6 | 69.8 | 74.7 | 68.6 |
| LVNet (Park et al., 2024) [2] | GPT-4o | 68.2 | 61.1 | 65.5 | 75.0 | 81.5 | 72.9 |
| *Based on Open-source Captioners and Proprietary LLMs* | | | | | | | |
| ProViQ (Choudhury et al., 2023) | GPT-3.5 | 57.1 | - | - | - | - | 64.6 |
| LLoVi (Zhang et al., 2023a) | GPT-3.5 | 57.6 | 50.3 | - | - | - | - |
| MoReVQA (Min et al., 2024) | PaLM-2 | - | 51.7 | 64.6 | 70.2 | - | 69.2 |
| Vamos (Wang et al., 2023c) | GPT-4 | 51.2 | 48.3 | - | - | - | - |
| LLoVi (Zhang et al., 2023a) | GPT-4 | 61.2 | - | 61.0 | 69.5 | 75.6 | 67.7 |
| VideoAgent (Wang et al., 2024c) | GPT-4 | 60.2 | 54.1 | 64.5 | 72.7 | 81.1 | 71.3 |
| VideoAgent (Fan et al., 2024) | GPT-4 | 62.8 | 60.2 | - | - | - | - |
| LifelongMemory (Wang et al., 2024g) [3] | GPT-4 | 64.1 | 58.6 | - | - | - | - |
| VIDEOTREE (Ours) | GPT-4 | **66.2** | **61.1** | **70.6** | **76.5** | **83.9** | **75.6** |

the sake of making a fair comparison, we also show VIDEOTREE's ability using open-source LLM in Tab. 4, where we obtain an 4.8% improvement on the EgoSchema subset. These results showcase the effectiveness of VIDEOTREE compared with existing training-free methods. Moreover, VIDEOTREE is also more efficient: we show analyses measuring the number of captions in Fig. 3 and inference time in Tab. 3, where VIDEOTREE is more efficient than relevant baselines.

**Evaluating on Very Long Videos.** To further highlight the strength of our approach on longer videos, we have included results on Video-MME (Fu et al., 2024a)'s long split, which contains a diverse set of very long videos (up to 1 hour, with an average of 44 minutes). We compare our training-free method with two types of models, including proprietary MLLMs (OpenAI, 2023a; 2024; et al., 2024) and open-source MLLM (Zhang et al., 2024a; Fu et al., 2024b; Chen et al., 2023b; 2024; Wang et al., 2024d; Zhang et al., 2024b; Wang et al., 2024b), both of which are trained on large-scale video(image) data. As shown in Tab. 2, compared to proprietary MLLMs, VIDEOTREE outperforms the strong GPT-4V (OpenAI, 2023a) model by 0.7% but still has a gap with the powerful long-context proprietary MLLMs (GPT-4o (OpenAI, 2024), Gemini 1.5 Pro (et al., 2024)). When comparing to open-source MLLMs that were extensively trained on video data, our training-free VIDEOTREE method outperforms a number of strong MLLMs VIDEOTREE including ViLA-1.5-40B (Lin et al., 2023c), Intern-VL2 (Chen et al., 2024). In summary, on very long videos, VIDEOTREE achieves strong performance without any additional training on video data.

## 5.2 ANALYSIS

Below, we provide a detailed analysis of VIDEOTREE framework. All quantitative analyses are conducted on the validation subset of the EgoSchema dataset. First, we analyze the trade-off between efficiency and effectiveness, showing that our method has better efficiency *and* performance across all settings compared to existing methods. We then provide an comprehensive ablation study for different design choice of VIDEOTREE (additional ablation study in Appendix Sec. C). Finally, we

---

[1] We de-emphasize the EgoSchema results of LangRepo since it predicts the answers via a log-likelihood classifier rather than generation, making it different from all other methods (including VIDEOTREE). We provide a comparison using the same classifier and LLM in Tab. 4 and show 4.8% improvements under same settings.

[2] For fair comparison, we de-emphasize methods that use a much stronger MLLM (GPT-4o) as both the captioner and the LLM.

[3] Reproduced results, implementation details in Sec. E

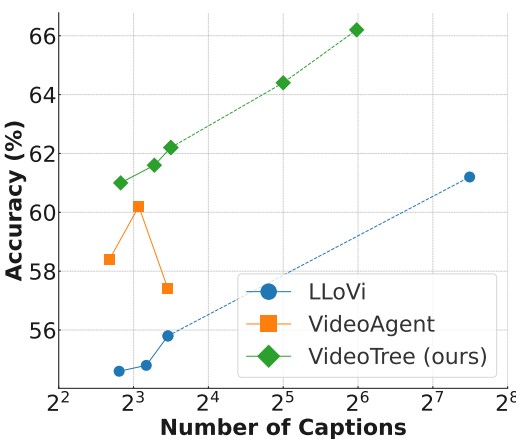

Figure 3: Ablating the number of captions. Given approximately the same number of frames, VIDEOTREE substantially outperforms LLoVi and VideoAgent. VIDEOTREE's hierarchical nature also allows it to generalize better to more frames and obtain better overall performance.

Table 2: Video-MME long split results. Our training-free VIDEOTREE method outperforms a strong proprietary MLLM (GPT-4V) and open-source MLLMs (e.g. ViLA-1.5-40B) which are extensively trained on videos.

| Method | Accuracy |
|---|---|
| *Proprietary MLLM* | |
| GPT-4V | 53.5 |
| GPT-4o | 65.3 |
| Gemini 1.5 Pro | 67.4 |
| *Open-Source MLLM* | |
| LongVA | 46.2 |
| VITA | 48.6 |
| InternVL2 | 52.6 |
| VILA-1.5-40B | 53.8 |
| LLaVA-NeXT-Video-72B | 61.5 |
| Qwen2-VL-72B | 62.2 |
| *Training-free Approach* | |
| VIDEOTREE (Ours) | 54.2 |

Table 3: Efficiency-Effectiveness comparison between LLoVi and our approach. We benchmark the time cost of VIDEOTREE and LLoVi (Zhang et al., 2023a), split into seconds spend in frame captioning, extracting keyframes, performing QA, and also report overall time. Using only 33% inference time, VIDEOTREE(fast) already achieves both better performance compared to LLoVi(best).

| Method | Captions | Captioner (s) | Keyfr. (s) | QA (s) | Overall (s) | Acc. |
|---|---|---|---|---|---|---|
| LLoVi-fast | 16 | 2.0 | 0 | 1.9 | **3.9** | 57.8 |
| LLoVi-best | 180 | 22.4 | 0 | 2.4 | 24.8 | 61.2 |
| VIDEOTREE-fast | **13.6** | **1.6** | 4.4 | **1.8** | 7.8 | 63.6 |
| VIDEOTREE-best | 62.4 | 7.8 | 10.2 | 2.1 | 20.1 | **66.2** |

visualize the tree representation from VIDEOTREE and show the clusters VIDEOTREE chooses to expand, qualitatively supporting its quantitative gains.

### 5.2.1 EFFICIENCY-EFFECTIVENESS ANALYSIS

In Tab. 3, we show the efficiency-effectiveness trade-off of our approach compared to existing methods. Specifically, we compare VIDEOTREE with LLoVi (Zhang et al., 2023a) using the same GPT-4 model as LLM (and same captioner). Comparing to the best model, LLoVi, VIDEOTREE-fast (which uses fewer frames by changing the hyper-parameters) achieves a 2.4% improvement on the EgoSchema subset with only 33% the time cost. Moreover, our best model obtains a 5.0% improvement with less overall inference time compared to both LLoVi models. Profiling the inference time spent in different modules (including frame captioning, extracting keyframes/caption summarization, performing QA), we find that our hierarchical keyframe selection consumes a reasonable amount of time while significantly reducing the time cost in the captioning stage and boosting long video understanding performance. We also provide an ablation of average LLM calls and compared with VideoAgent (Wang et al., 2024c) in Tab. 8 showing that VIDEOTREE requires fewer LLM calls while having better performance. These results show that VIDEOTREE has better effectiveness and efficiency compared to the existing method.

### 5.2.2 ABLATION STUDY

In this section, we conduct ablate different parts of VIDEOTREE on the EgoSchema subset. We ablate four features: Number of captions, visual encoder choice, applying open-source LLM and different

Table 4: Accuracy on the EgoSchema subset when using open-source LLM Reasoners and log-likelihood classifier. VIDEOTREE obtains better performance with less inference time on both 7B and 12B LLMs comparing to the LangRepo baseline (Kahatapitiya et al., 2024).

| Method | LLM | # Caption | Acc. | Inf Time (s) |
|---|---|---|---|---|
| LLoVi | Mistral-7B | 180 | 50.8 | - |
| LangRepo | Mistral-7B | 180 | 60.8 | 87.2 |
| VIDEOTREE (ours) | Mistral-7B | 32 | **63.0** | **24.3** |
| LangRepo | Mixtral-8×7B (12B) | 180 | 66.2 | 162.1 |
| VIDEOTREE (ours) | Mixtral-8×7B (12B) | 32 | **71.0** | **50.3** |

VIDEOTREE components. We include more extensive ablations (including hyper-parameter analysis and the design choices of captioner/LLM) in Appendix Sec. C.

**Number of Captions.** In Fig. 3, we compare VIDEOTREE with existing methods under different caption settings. Under similar frame caption settings (7, 9, 11), VIDEOTREE outperforms LLoVi (Zhang et al., 2023a) and VideoAgent (Wang et al., 2024c) by 6.5% and 2.0% on average accuracy across all three settings. Moreover, unlike the non-hierarchical VideoAgent baseline, which suffers from performance degradation after 11 frames, our method continues improving, generalizing to 62.4 frames and achieving 6% better accuracy in terms of best performance. This result highlight the importance of VIDEOTREE's hierarchical nature.

**Open-source LLM Reasoner.** To validate the effectiveness of VIDEOTREE with open-source LLM reasoners (rather than GPT4), in Tab. 4, we report the performance of VIDEOTREE using 7B and 12B versions of the Mistral model (Jiang et al., 2023; 2024) as the LLM reasoner. We compare with LLoVi (Zhang et al., 2023a) and LangRepo (Kahatapitiya et al., 2024). For a maximally fair comparison, we follow LangRepo's evaluation pipeline, using a log-likelihood classifier that scores all options and takes the highest-scoring one. VIDEOTREE substantially outperforms the baseline approaches on both 7B and 12B Mistral models while only requiring 20% of the frame captions. Specifically, compared to LangRepo, which uses complex textual summarization modules, VIDEOTREE achieves 2.2% and 4.8% better EgoSchema subset performance while using about 72.5% and 69.0% less inference time on Mistral 7B and 12B LLM, respectively. These results confirm that VIDEOTREE's effectiveness and efficiency translates to open-source reasoners as well.

Table 5: Ablation studies of VIDEOTREE on visual encoder and different components.

(a) Visual encoder ablation.

| Visual Encoder | Params | Method | Acc. |
|---|---|---|---|
| OpenCLIP-ViT-B | 88M | VideoAgent | – |
| | | VIDEOTREE | 66.0 |
| OpenCLIP-ViT-G | 1B | VideoAgent | 59.2 |
| | | VIDEOTREE | **66.2** |
| EVA-CLIP-8B | 8B | VideoAgent | 59.4 |
| | | VIDEOTREE | **66.2** |

(b) Effect of different VIDEOTREE components. Both Adaptive Breadth Expansion and Depth Expansion modules contribute significantly to the effectiveness of VIDEOTREE.

| Module | ES Acc. |
|---|---|
| VIDEOTREE | 66.2 |
| - Depth Expansion | 64.4 |
| - Adaptive Breadth Expansion | 61.2 |

**Visual Encoder.** In Tab. 5a, we study the effect of the visual encoder used in the visual clustering operation. We report the results of VIDEOTREE on three different scales of visual encoder: OpenCLIP-B, OpenCLIP-G (Ilharco et al., 2021) and EVA-CLIP-8B (Sun et al., 2024) and compare to VideoAgent (Wang et al., 2024c) [2]. VIDEOTREE outperforms VideoAgent by an average of 6.9% across both encoders. Comparing different visual encoders ranging from 88M to 8B parameters, we see only a marginal drop in performance for VIDEOTREE as the visual encoders decrease in size, indicating that our approach generalizes well to much smaller vision encoders (i.e. only a 0.2% drop when going from 8B to 88M), making the model more efficient while maintaining strong performance.

**VIDEOTREE Components.** In Tab. 5b, we report the effectiveness of the different components in VIDEOTREE. Specifically, removing the depth expansion module brings a 1.8% drop in performance,

---

[2]Note that VideoAgent only report results on OpenCLIP-ViT-G (1B) and EVA-CLIP-8B.

showing the importance of the hierarchical design of VIDEOTREE. Furthermore, removing the adaptive breadth expansion module brings another 3.2% performance decrease, verifying the effectiveness of the adaptive nature of VIDEOTREE.

### 5.2.3 QUALITATIVE ANALYSIS

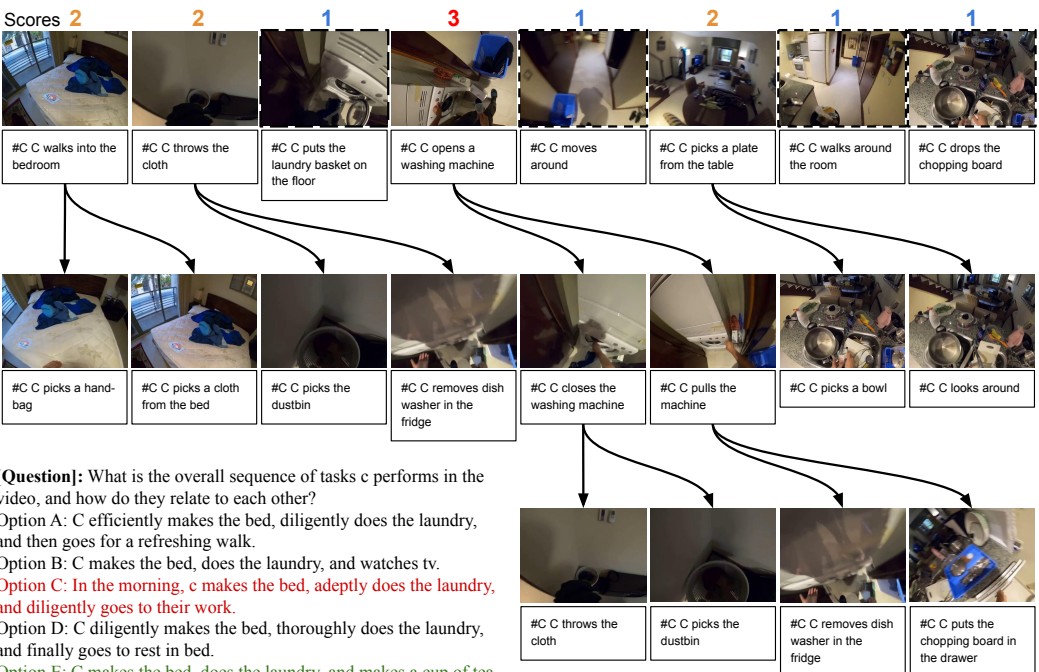

[Question]: What is the overall sequence of tasks c performs in the video, and how do they relate to each other?
Option A: C efficiently makes the bed, diligently does the laundry, and then goes for a refreshing walk.
Option B: C makes the bed, does the laundry, and watches tv.
Option C: In the morning, c makes the bed, adeptly does the laundry, and diligently goes to their work.
Option D: C diligently makes the bed, thoroughly does the laundry, and finally goes to rest in bed.
Option E: C makes the bed, does the laundry, and makes a cup of tea.

Figure 4: Qualitative examples of VIDEOTREE. Red options are answered wrongly with uniformly sampled 32 frames. Green options are answered correctly with VIDEOTREE. Best viewed in color.

In Figure 4, we visualize qualitative results from VIDEOTREE. Specifically, we show the keyframes and their captions extracted by our adaptive tree representation given a video query. This example is drawn from EgoSchema, and shows the query format, which consists of a query and multiple-choice answers. With the proposed VIDEOTREE strategy, we split a complex multi-scene video (*e.g.cleaning house across rooms*) into several key scenes via visual clustering and determine the most query-relevant scene via the relevance score. We then obtain more fine-grained visual cues by descending into each relevant cluster (Levels 2 and 3 in Figure 4). For example *"C opens a washing machine"* is deemed highly relevant to the question, which asks about the sequence of events. At the same time, frames like *"C moves around"* are deemed irrelevant to the query and not expanded. In the end, VIDEOTREE shows a dynamic ability to select relevant segments and answer the given question correctly with only 50% of the baseline's 32 input captions. The LLoVi (fixed uniformly sampling) fails to correctly answer the question, sampling a large number of redundant and irrelevant frames. We also provide additional qualitative results and failure case visualizations in Appendix Sec. F.

## 6 CONCLUSION

In this work, we proposed VIDEOTREE, an adaptive and hierarchical framework for LLM reasoning over long-form videos. VIDEOTREE adaptively extracts query-relevant keyframes from the video input in a coarse-to-fine manner and organizes them into a hierarchical representation, enabling the LLM to effectively handle complex queries. VIDEOTREE resulted in strong performance on three popular datasets (EgoSchema, NExT-QA, and Video-MME), while also improving efficiency by reducing the inference time and LLM calls. We analyzed the role of the adaptive cluster selection we implement in VIDEOTREE, finding that it is crucial to strong performance. In our qualitative analysis, we showed that given a complex multi-scene video and its query, VIDEOTREE is capable of extracting key scenes and zooming into more detailed information that is highly related to the query.

ETHICS STATEMENT

The intended use of VIDEOTREE is to answer questions about long-form videos. This does not have any particular potential for misuse beyond the general potential for AI technology to be used in harmful ways. Because it is based on VLM captioning and LLMs answering questions from captions, VIDEOTREE has the potential to hallucinate (both in the captioning stage and the QA stage). Note that potential is shared both with other caption-then-LLM approaches (Zhang et al., 2023a; Wang et al., 2024c; Kahatapitiya et al., 2024) and end-to-end VLM-based approaches(Zhang et al., 2024b; Lin et al., 2023c), which are also prone to hallucination (Ji et al., 2023). The fact that VIDEOTREE answers questions based on coarse-to-fine captions that are tied to particular frames in a video makes it more interpretable than black-box methods, mitigating the potential for hallucinations that mislead users, since users could potentially examine the sampled frames as we have done in Fig. 4.

REPRODUCIBILITY STATEMENT

To maximize reproducibility, we have included our code in the supplementary material. Proprietary LLMs present reproducibility concerns, since models are updated and taken offline over time. To counteract this, in addition to our results on proprietary LLMs, we have included results using only open-source LLMs and captioners in Tab. 4. Finally, we report all of our hyperparameter settings and model details in Sec. E.

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

APPENDIX

In this Appendix, we present the following:

- Limitations and Broader Impact (Sec. A).
- Additional quantitative results (Sec. B).
- Additional ablation study for VIDEOTREE framework (Sec. C).
- The detailed algorithm for VIDEOTREE (Sec. D).
- Additional implementation details (Sec. E).
- Additional qualitative analysis (Sec. F).
- License information (Sec. G).

## A  LIMITATIONS AND BROADER IMPACT

**Limitations.**   Like all LLM-based video-reasoning systems (including dense sampling) our method is limited by the ability of the captioner to accurately capture the contents of sampled frames. However, our method's modular nature means that as captioners improve, we can easily include them into the VIDEOTREE framework; similarly, we can use increasingly strong LLMs as the reasoning backbone of VIDEOTREE. While VIDEOTREE is training-free, it includes a small number of hyperparameters. In Sec. C, we ablate these hyperparameters, showing that VIDEOTREE outperforms the uniform-sampling baseline regardless of the choice of max depth and branch width. Thus, while better hyperparameters can benefit the method, even with sub-optimal settings VIDEOTREE outperforms the uniform baseline.

**Broader Impact.**   Our results indicate that we have the best of both worlds: improved accuracy *and* improved efficiency. Given the importance of long video reasoning tasks, improving accuracy has obvious broader implications for building more usable video reasoning systems, which could contribute to a wide variety of positive applications. Efficiency improvements also contribute to the applicability of long video systems, as reducing latency and computational cost can speed up adoption. Furthermore, since both VLM captioners and LLM reasoners generally improve with increased scale, reducing the number of calls to them will become increasingly important; we expect the efficiency benefits coming from our method to play an even larger role in the future. Our work does not have any particularly relevant potential for negative applications or misuse beyond the standard caveats that apply to all machine learning systems.

## B  ADDITIONAL QUANTITATIVE RESULTS

### B.1  COMPARISON WITH ADVANCED VIDEOLLMS ON EGOSCHEMA AND NEXT-QA.

In Tab. 6, we compare VIDEOTREE with advanced VideoLLMs (Wang et al., 2022b; 2024f; Cheng et al., 2024; Wang et al., 2024a) on EgoSchema and NExT-QA benchmarks. Without any video-specific training, VIDEOTREE gets comparable performance on EgoSchema fullset and slightly worse results on Next-QA results, comparison with the methods was trained on large-scale video data and massive GPU hours.

### B.2  INTENTQA RESULTS

In this section, we report the IntentQA (Li et al., 2023) results of VIDEOTREE and compare with existing methods.

We first introduce IntentQA (Li et al., 2023), which contains 4,303 videos and 16K multiple-choice question-answer pairs focused on reasoning about people's intent in the video. We perform a zero-shot evaluation on the test set containing 2K questions. The videos are more than 44s in average length. We compare our methods with both training-free (Kahatapitiya et al., 2024; Zhang et al., 2023a; Yu et al., 2024a; Kim et al., 2024) and fine-tuned baseline (Wang et al., 2023c; Xiao et al., 2022b).

As shown in Tab. 7, our training-free VIDEOTREE approach achieves 66.9% zero-shot accuracy on the test set, surpassing the existing training-free approaches LLoVi (Zhang et al., 2023a) with 2.7% improvements and even closing the gap with finetuned method Vamos (Wang et al., 2023c). This result shows that VIDEOTREE improves performance in answering questions about intent, which is challenging since intent understanding (Li et al., 2023) requires the model to understand the various video contexts, including the immediate communicative context, the shared experience, and the commonsense.

Table 6: Comparision with advanced VideoLLMs on EgoSchema and NExT-QA.

| Method | ES | NExT-QA |
|---|---|---|
| InternVideo | 32.1 | - |
| Tarsier | 61.7 | 79.2 |
| VideoChat2 | 60.2 | 61.7 |
| VideoLLaMA 2 | 63.9 | - |
| *Traning-free Methods* | | |
| VIDEOTREE (ours) | 61.1 | 75.6 |

Table 7: IntentQA Results

| Method | Accuracy |
|---|---|
| *Fine-tuned Method* | |
| VGT | 51.3 |
| Vamos | 68.5 |
| *Traning-free Methods* | |
| LangRepo | 59.1 |
| SeViLA | 60.9 |
| LLoVi | 64.0 |
| IG-VLM | 64.2 |
| VIDEOTREE (ours) | 66.9 |

## C    ADDITIONAL ABLATION STUDY

In this section, we report additional ablation studies for VIDEOTREE framework. First, we ablate the LLM calls for our method. Then, we show the effect of the different hyperparameter settings for VIDEOTREE. Finally, we analyze the effect of different VLM/LLM designs for VIDEOTREE.

**LLM Calls.** In Tab. 8, we report the number of average LLM calls of VIDEOTREE and compare with VideoAgent (Wang et al., 2024c).VIDEOTREE achieves better results on much less LLM calls under similar caption numbers (only about 30% LLM calls are needed). This is due to the adaptive and hierarchical structure of VIDEOTREE which could extracts more keyframes faster instead of searching one frame a time. This results highlight the advantage of the hierarchical nature of VIDEOTREE in both efficiency and effectiveness, comparing to the non-hierarchical approaches.

**Hyperparameter Analysis.**

In Tab. 9, we study the effect of the branch width of the tree-based representation for the VIDEOTREE. The best performance is obtained when the branch width is set to 4. As with depth, excessive branch width reduces the VIDEOTREE performance due to the information overwhelming to the LLM; however, even with the worst branch width setting, VIDEOTREE still outperforms the baseline.

Table 8: The comparison of average LLM calls for VIDEOTREE and VideoAgent (Wang et al., 2024c) (estimated) under similar frame settings on EgoSchema subset. Results show that VIDEOTREE achieves better results on much less LLM calls.

| Caption Number | Avg LLM Calls | ES Subset Acc |
|---|---|---|
| *VideoAgent* | | |
| 6.4 | 10.2 | 58.4 |
| 8.4 | 10.2 | 60.2 |
| 11.0 | 9 | 57.4 |
| VIDEOTREE *(ours)* | | |
| 7.1 | **2.3** | **61.0** |
| 9.7 | **2.5** | **61.6** |
| 11.3 | **2.8** | **62.2** |

Table 9: The effect of different settings for branch width of VIDEOTREE. When the branch width is set to 4, VIDEOTREE achieves the best performance on the EgoSchema subset. Reducing the branch width makes the model more efficient while retaining performance, outperforming all existing approaches.

| Branch Width | ES Acc↑ | #Frame↓ |
|---|---|---|
| 2 | 64.4 | 43.5 |
| 3 | 65.0 | 54.6 |
| 4 | **66.2** | 62.4 |
| 5 | 64.2 | 72.5 |
| Uniform Baseline | 61.2 | 180 |

In Tab. 10, we study the effect of the max breadth of the adaptive tree-based representation. The results indicate that even with a smaller max tree breadth, VIDEOTREE achieves good performance while using much fewer frames. Increasing the breadth generally increases performance, with the best performance when the max breadth is set to 32. However, having an excessive max breadth leads to worse results, suggesting that incorporating too much information in the adaptive tree-based representation limits the LLM reasoning ability. This links back to the intuition of having an efficient representation for the LLM reasoning over long videos.

In Tab. 11, we study the effect of the threshold on the number of highly relevant clusters, which controls the iterative process of the adaptive breadth expansion process. The best performance is obtained when the branch threshold is set to 4. Reducing the threshold improves the efficiency while retaining strong performance compared to the baseline results.

Table 10: The effect of different settings for the max breadth of the first level of the tree. Results show that when the max breadth is set to 32, VIDEOTREE obtains the best performance. Reducing the max breadth improves efficiency while retaining performance.

| Max Breadth | ES Acc | #Frame |
|---|---|---|
| 8 | 63.0 | 26.9 |
| 16 | 64.0 | 49.0 |
| 32 | **66.2** | 62.4 |
| 64 | 63.2 | 94.6 |
| Uniform Baseline | 61.2 | 180 |

Table 11: The effect of different settings for the threshold on the number of highly relevant clusters. Results show that when the threshold is set to 4, VIDEOTREE obtains the best performance. Reducing the threshold improves efficiency while retaining performance.

| Max Breadth | ES Acc | #Frame |
|---|---|---|
| 2 | 63.6 | 13.9 |
| 3 | 64.4 | 32.2 |
| 4 | **66.2** | 62.4 |
| 5 | 64.8 | 79.2 |
| Uniform Baseline | 61.2 | 180 |

**VLM Captioner.** In Tab. 12, we compares the performance of the best captioner (LaViLA for EgoSchema and CogAgent for NExT-QA) with using a LLaVA-1.6-7B (Liu et al., 2024) captioner everywhere. We observe a comparable performance on NExT-QA compared with the best captioner, while still outperforming all other existing methods in Tab. 1. We also observe a drop in performance on the EgoSchema subset while using LLaVA-1.6 captioner, this is likely due to a lack of egocentric data during LLaVA training, which is needed for strong performance on EgoSchema. In the future, we would like to see strong unified captioner that operate well across datasets; these would fit seamlessly into the VIDEOTREE framework, further boosting the performance.

**LLM Reasoner.** We ablate the design choice of captioner and LLM for the VIDEOTREE framework in Tab. 13. With a better LLM, VIDEOTREE can achieve better performance on long video understanding tasks, indicating the potential VIDEOTREE to improve as its modules become more advanced. Notably, our GPT-3.5 variant substantially outperforms existing methods with the same LLM and standard QA prompts (VideoAgent (Wang et al., 2024c) 48.8%, LLoVi (Zhang et al., 2023a) 51.8%), achieving 57.6% accuracy on EgoSchema subset.

Table 12: Comparing accuracy with VIDEOTREE using the same captioner throughout (LLaVA1.6-7B) and best captioner for each benchmarks.

| Captioner | EgoSchema Sub | NExT-QA |
|---|---|---|
| LLaVA-1.6-7B | 59.2 | 73.6 |
| Best Model | **66.2** | **75.6** |

Table 13: The effect of different design choices of the LLM Reasoner for VIDEOTREE.

| Method | LLM | ES Acc |
|---|---|---|
| LLoVi | GPT-3.5 | 51.2 |
| VideoAgent | GPT-3.5 | 48.8 |
| VIDEOTREE (Ours) | GPT-3.5 | **57.6** |
| LLoVi | GPT-4 | 61.2 |
| VideoAgent | GPT-4 | 60.2 |
| VIDEOTREE (Ours) | GPT-4 | **66.2** |

## D    DETAILED ALGORITHM

In Algorithm 1, we present the algorithm behind VIDEOTREE.

---

**Algorithm 1** VIDEOTREE

---

**Require:** Video frames $V$, query $Q$, number of clusters $k$, threshold for the number of high-relevance cluster $rele\_num\_thresh$, maximum number of clusters allowed $max\_breadth$, branch width $w$, visual encoder $E$, LLM $F_{llm}$, captioner $F_{vlm}$, cluster information $C$, relevance score $R$, tree-based video representation $Tree$

1:   $k \leftarrow k\_init$
2: **while** $k \leq max\_breadth$ **do**                 ▷ Adaptive breadth expansion
3:      $C \leftarrow$ VisualClustering($E, V, k$)
4:      $Cap \leftarrow$ ClusterCaptioning($F_{vlm}, V, C$)
5:      $R \leftarrow$ RelevanceScoring($F_{llm}, C, Q, Cap$)
6:      **if** count($r \in R \mid r = high$) $\geq rele\_num\_thresh$ **then**
7:           $Tree \leftarrow Tree.append(C)$             ▷ First level of VIDEOTREE
8:           **break**
9:      **end if**
10:     $k \leftarrow k * 2$
11: **end while**
12: **for** $C_i \in C$ **do**                  ▷ Relevance-guided depth expansion
13:     $\hat{C}_i \leftarrow$ DepthExpansion($E, C_i, R_i, w$)
14:     $Tree \leftarrow Tree.append(\hat{C}_i)$           ▷ Adding hierarchy of VIDEOTREE
15: **end for**
16: $Cap \leftarrow$ GetCaptions($F_{vlm}, V, Tree$)                ▷ LLM Reasoning
17: $pred\_answer \leftarrow$ LLMReasoning($F_{llm}, Cap, Q$)
18: **return** $pred\_answer$

---

## E    ADDITIONAL IMPLEMENTATION DETAILS

**Additional VIDEOTREE Implementation Details.** For clustering, we use the `kmeans_pytorch` library. The hyper-parameter setting for $max\_breadth$, $max\_depth$, $branch\_width$ and $rele\_num\_thresh$ on the EgoSchema and Video-MME benchmark is 32, 3, 4 and 4 and for NExT-QA, we set the hyper-parameter as 8, 3, 2, and 3.

**Lifelong Memory Reproduce Details.** In Tab. 1, we report the main results of LifelongMemory (Wang et al., 2024g) which is lower than the number than they reported in their paper. Here, we

Table 14: **VIDEOTREE with relevance scoring prompt on EgoSchema.**

---

**User**

You are presented with a textual description of a first view video clip, it consists of about `caption_number` frame captions sparsely sampled from the video (#C means the first person view, and #O indicates another). The ultimate goal is to answer a question related to this video, choosing the correct option out of five possible answers.

It is crucial that you imagine the visual scene as vividly as possible to enhance the accuracy of your response. After selecting your answer, rate your confidence level in this choice on a scale from 1 to 100, where 1 indicates low confidence and 100 signifies high confidence. Please provide a concise one-sentence explanation for your chosen answer. If you are uncertain about the correct option, select the one that seems closest to being correct. Meanwhile, could you provide a relevance score for each frame caption to evaluate their relevance with the query-answering process. The score is between 1,2,3, where 1 indicates low relevance and 3 signifies high relevance. Please return the relevance score in the format of a list of `caption_number` scores.
Examples: `Examples`
Description: `Captions`
Question: `Question`
Options: A: `Option-A`. B: `Option-B`. C: `Option-C`. D: `Option-D`. E: `Option-E`.
The prediction, explanation, confidence and frame relevance are (please response in the format of 'prediction:, explanation:, confidence:, frame relevance:')

---

**Assistant**

prediction, explanation, confidence, frame relevance

---

introduce our reproduce process in detail. For captions, since LifelongMemory authors do not provide the exact caption data/path, we directly utilize the same captioner from VIDEOTREE method and all other existing works (LaViLA) and extract the captions by 0.5FPS according to LifelongMemory paper. We then use their code to run the experiments on EgoSchema, however, the results are low and we observed a low success rate of the QA process (only about 80% success samples). We then update their output format/process code, which boost performance by about 10% and get the results in Tab. 1, but still lower than their paper results. Thus, for fair comparison, we directly reported the reproduced results.

**Prompt Details.** We provide detailed prompts for the relevance scoring prompt in Tab. 14 and LLM reasoning prompt in Tab. 15 on the EgoSchema benchmark.

**Experiments Compute Resources.** All experiments are conducted on 4 (or less) NVIDIA-A6000 GPU and Azure Cloud APIs (for OpenAI models). The minimal GPU memory requirement is 24GB.

## F    ADDITIONAL QUALITATIVE ANALYSIS

**Additional Visualization.**   In Fig. 5 we show another visualization from VIDEOTREE. Here, VIDEOTREE localizes a single key activity (embroidering a cloth) taking place in the video and dynamically expands its constituent frames to answer the question correctly using a minimal number of frames.

**Failure Case.** We provide the qualitative visualization in Fig. 6. We find the failure was due to the following factors: a. The video had little scene change and multiple similar repeated actions (washing dishes). b. As a result, when VIDEOTREE explores to more fine-grained details, the captioners give

Table 15: **VIDEOTREE with LLM reasoning prompt on EgoSchema.**

---

**User**

You are presented with a textual description of a first view video clip, it consists of frame captions sparsely sampled from the video (#C means the first person view, and #O indicates another). The ultimate goal is to answer a question related to this video, choosing the correct option out of five possible answers.

It is crucial that you imagine the visual scene as vividly as possible to enhance the accuracy of your response. After selecting your answer, rate your confidence level in this choice on a scale from 1 to 100, where 1 indicates low confidence and 100 signifies high confidence. Please provide a concise one-sentence explanation for your chosen answer. If you are uncertain about the correct option, select the one that seems closest to being correct.

Examples: Examples
Description: Captions
Question: Question
Options: A: Option-A. B: Option-B. C: Option-C. D: Option-D. E: Option-E.
The prediction, explanation, and confidence is (please response in the format of 'prediction:, explanation: ,confidence:')

---

**Assistant**

prediction, explanation, confidence

---

detailed (with a bit of a hallucination) description, which misses the correct keyword (dish) in its selected captions. c. With stronger captioners, this failure case could be resolved by our framework.

## G    LICENSE

We will make our code and models publicly accessible. We use standard licenses from the community and provide the following links to the licenses for the datasets, codes, and models that we used in this paper.

**LLoVi:** MIT

**LifelongMemory:** MIT

**NExT-QA:** MIT

**IntentQA:** IntentQA

**EgoSchema:** Ego4D license

**Kmeans-pytorch:** MIT

**PyTorch:** BSD-style

**Huggingface Transformers:** Apache

**Torchvision:** BSD 3-Clause

**SKLearn:** BSD 3-Clause

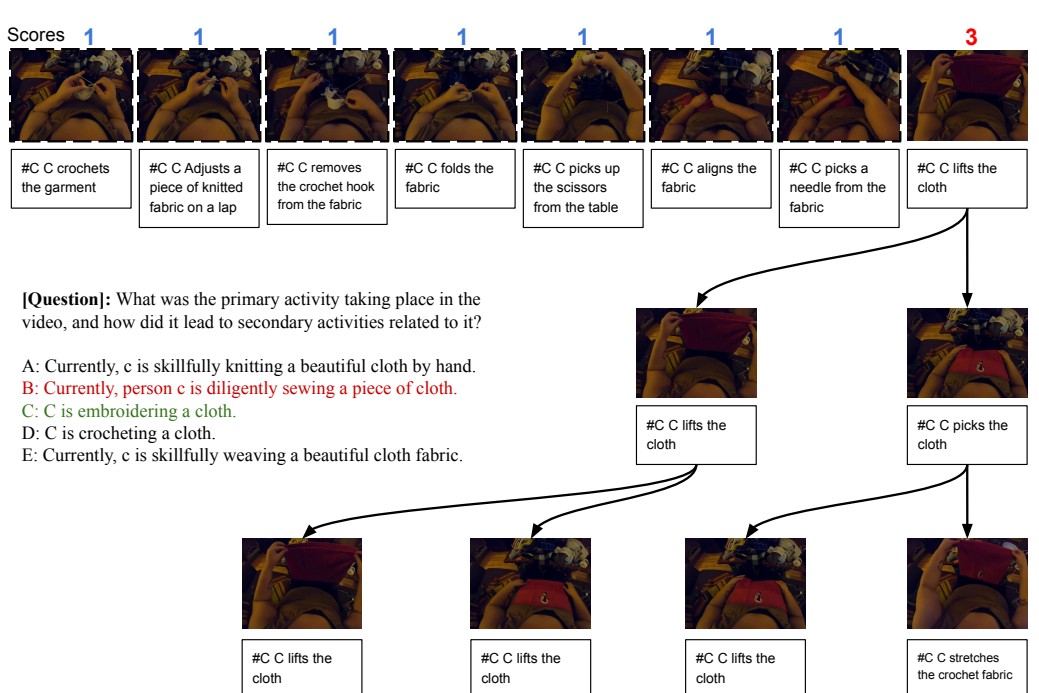

Figure 5: Qualitative examples of VIDEOTREE keyframes and captions selection. Red options are answered wrongly with uniformly sampled frames. Green options are answered correctly by VIDEOTREE. Best viewed in color.

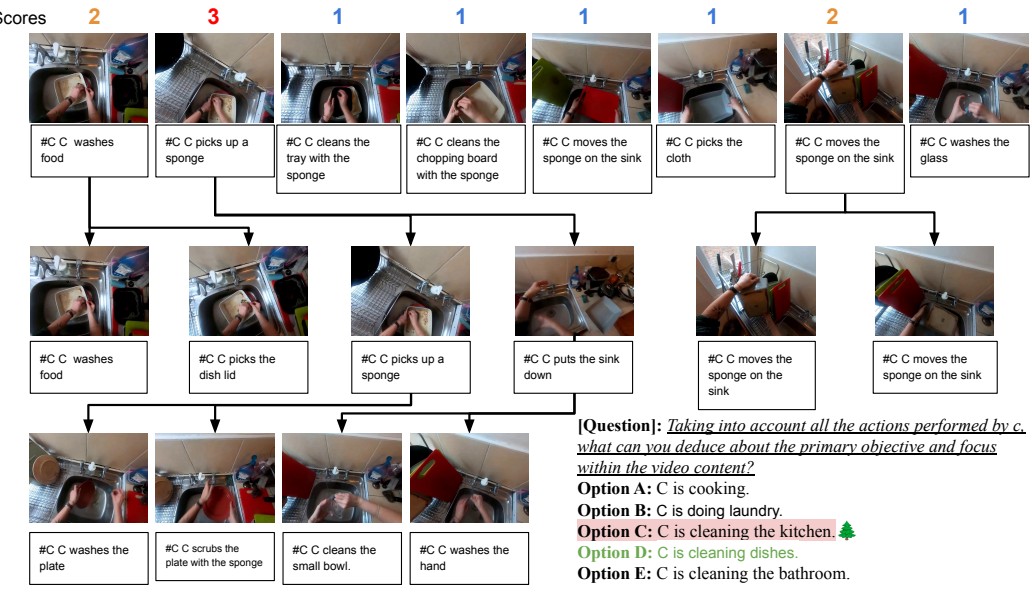

Figure 6: Failure Case of VIDEOTREE.

