# OpenReview forum: "VideoTree: Adaptive Tree-based Video Representation for LLM Reasoning on Long Videos"
_ICLR.cc/2025/Conference — ICLR 2025 Conference Withdrawn Submission_

### Official Review · Reviewer_FedZ · 2024-10-23

**Soundness:** 3
**Presentation:** 4
**Contribution:** 2
**Rating:** 5
**Confidence:** 4

**Summary:**

The paper introduces a training-free framework for long-form video understanding that improves reasoning efficiency and accuracy by constructing a hierarchical query-relevant video representation. Its key contributions are:

1.  *Query-Adaptive Frame Selection*: It dynamically selects keyframes based on their relevance to the query, minimizing the inclusion of redundant or irrelevant information, improving both speed and accuracy. This is done via the following steps:
    - Perform an **Adaptive Breadth Expansion** on the video frames which clusters them based on their similarities, captions each clusters and then score them based on their relevance to the query.

    - Deepen the tree via a **Relevance-guided Depth Expansion** which sub-clusters the most relevant clusters allowing more fine-grained representations of the query-relevant parts of the video.
2. *Training-Free Efficiency*: Unlike many methods that require extensive video-specific training, this method performs well without training. It outperforms existing training-free models on EgoSchema, NExT-QA, and Video-MME, achieving superior performance and faster inference times.

**Strengths:**

- The paper is well-structured and the flow of idea is easy to follow.
- By dynamically selecting only the most relevant keyframes through Adaptive Breadth Expansion and Relevance-guided Depth Expansion, the method reduces the noise caused by irrelevant or redundant frames. This leads to more accurate reasoning over long videos and avoids overwhelming the model with unnecessary data.
- The method is computationally efficient due to its sparse selection of keyframes and hierarchical tree structure, which reduces the need for dense frame processing.

**Weaknesses:**

- *Dependence on the initial keyframe selection and the value of k*: During the Adaptive Breadth Expansion, the initial clustering might miscluster some frames, and the error might propagate through the whole tree. How is this accounted for in the process? Is VIDEOTREE's performance highly dependent on the number of clusters (k)? Could you include ablation experiments to test for that?
- *Many moving parts*: The paper thoughtfully aggregates several off-the-shelf models. Would the performance be bottlenecked by a suboptimal captioner or LLM reasoner?
- *Limited Novelty*: The main contribution of the paper is a new way to assemble a tree for long-form video understanding, which is very limited in scope given that the other important parts of the framework are off-the-shelf. Is it possible to extend it to other tasks?

**Questions:**

See weaknesses

---

### Official Review · Reviewer_2kd9 · 2024-11-02

**Soundness:** 3
**Presentation:** 3
**Contribution:** 2
**Rating:** 5
**Confidence:** 4

**Summary:**

This paper introduces VideoTree, a framework that offers a dynamic hierarchical video representation enabling LLMs to reason over long videos.

The method has 3 stages:

- The first stage—adaptive breath expansion—utilizes K-means to cluster visual features extracted from video frames, a captioner to obtain text descriptions, and an LLM to assign a relevance score for each cluster.
- These clusters are expanded according to their relevance scores in the next stage, with the most relevant clusters expanding into two-level trees.
- Finally, in the last stage, this tree is traversed top-to-bottom in temporal order to obtain a textual description of the video, which is then fed into the LLM alongside the query. Intuitively, tree representation aims to reduce the high redundancy of information in video data while preserving the fine-grained details that are relevant to the query.

The authors claim that VideoTree outperforms non-hierarchical methods such as VideoAgent and LLoVi on EgoSchema and NExT-QA benchmarks, obtaining a result comparable to LVNet, which capitalizes on the stronger GPT-4o backbone.

On Video-MME benchmark, which features long videos up to 1 hour in length, VideoTree slightly outperforms the proprietary GPT-4V model but comes up short against GPT-4o and Gemini 1.5 Pro.

Against 6 open-source MLLMs that were extensively trained on video data, VideoTree defeats 4 of them (Table 2) despite being a training-free approach.

**Strengths:**

Although hierarchical video representation existed before this paper, they operated in a bottom-up manner. In contrast, the proposed method improves efficiency and effectiveness by employing a top-down approach with dynamic depth. Thus, the method exhibits sufficient novelty. By surpassing the performance of previous training-free methods, the method attempts to demonstrate its significance (However, it falls short against state-of-the-art MLLMs, both proprietary and open-source.)

The paper explains the method clearly, with diagrams visualizing the information flow. The main text provides a concise overview, which is supplemented by the appendix that provides further details.

**Weaknesses:**

Throughout the paper, the authors claim that both redundancy and irrelevance of the information is harmful (e.g., in line 10, "Long-form video understanding has been a challenging task due to the high redundancy in video data and the abundance of query-irrelevant information.").  Although it looks easy to intuitively recognize the detrimental effect of irrelevant information, the claim about redundancy is not well-substantiated, especially in terms of empirical proof. In fact, one could argue that some of the results constitute empirical proof against this claim. Specifically, Figure 3 demonstrates that increasing the number of captions improves performance for both LLoVi and VideoTree.

Due to temporal redundancy in videos, more captions translate to more redundancy, which should be detrimental according to the authors, yet this seems to improve the accuracy. Moreover, the proposed method—despite filtering out irrelevant content—introduces more redundancy in a different form because of the tree structure. Given that reducing redundancy is a major motivation behind the proposed method, authors should justify their claim about redundancy, either by citing relevant work (if available) or providing empirical proof.

In line 30, the authors write "VIDEOTREE framework achieves better performance than the strong proprietary GPT-4V model and other MLLMs that were extensively trained on video data." I think this sentence seems to suggest VideoTree outperforms all tested MLLMs, which does not seem to be the case, as shown in Table 2. The authors should clarify this point by replacing "other MLLMs" with "many other MLLMs", for example. Overall, the authors should make it clear that the proposed method does not outperform the best proprietary and the best open-source MLLMs.

Although the fact that it can outperform numerous MLLMs despite being a training-free method is a technical feat, the paper doesn't explain the advantages of this method over an off-the-shelf MLLM. In other words, given that the training cost of MLLM has already been paid, why should we use a training-free approach? Therefore, more justification as to why this training-free method is preferable to using a pretrained MLLM would help emphasize the importance of the method. For instance, the authors could show that MLLMs require more computation and longer inference times.

Finally, the paper does not consider the possibilities for future work. The authors can address this by briefly mentioning some ideas in the conclusion section. This consideration can also improve the paper's position within the literature, thereby highlighting its significance.

Minor writing mistakes:
- Space before dot at line 128.
- Line 285: "recent-proposed" should be "recently proposed".

**Questions:**

1. In line 450, the authors write "...VideoAgent baseline, which suffers from performance degradation after 11 frames..." However, the x-axis in Figure 3 represents the number of captions. Did the authors mean "captions" instead of "frames"? Are they referring to the number of captioned frames?

2. Line 450: "our method continues improving, generalizing to 62.4 frames". What is the meaning of this fractional frame count? Assuming that they are referring to the number of captioned frames, the question still stands. Was this value averaged over video samples? Did this situation arise because some videos are shorter?

3. The caption of Table 8 states that VideoAgent's avg. LLM calls are estimated. Why weren't real values used? How were they estimated?

4. The prompts in Table 14 and 15 include queries about confidence. How are these confidence values used?

5. What is the reasoning behind the FPS choice (1 FPS for EgoSchema and NExT-QA, 0.125 for Video-MME)? How would the performance change if the FPS were changed?

**Details Of Ethics Concerns:**

-

---

### Official Review · Reviewer_6mvY · 2024-11-03

**Soundness:** 3
**Presentation:** 3
**Contribution:** 3
**Rating:** 5
**Confidence:** 5

**Summary:**

This paper introduces the VIDEOTREE, an adaptive and hierarchical framework for LLM reasoning over long-form videos. VIDEOTREE adaptively extracts query-relevant keyframes from the video input in a coarse-to-fine manner and organizes them into a hierarchical representation, enabling the LLM to effectively handle complex queries. Abundant experiments on three popular datasets EgoSchema, NExT-QA, and Video-MME shows the excellent performance of the VIDEOTREE.

**Strengths:**

1.	This paper is well written and easy to understand.
2.	The proposed training-free VIDEOTREE achieves better performance than the strong proprietary GPT-4V model and other MLLMs that were trained on video data on the long split of Video-MME benchmark.

**Weaknesses:**

1. The applicability of the proposed method is limited, as its effectiveness has only been verified on tasks such as multiple-choice questions. How about its performance on other video understanding-related tasks (open-ended VideoQA or text generation), such as action recognition, text-video localization, temporal reasoning tasks, and prediction-related tasks?
2. Although the authors claim to use coarse-to-fine hierarchical feature extraction, essentially, it still involves aggregation at the video frame level. This will prevent the model from effectively extracting fine-grained information within video frames, thereby limiting its performance on finer-grained video understanding tasks.
3. This algorithm requires multiple uses of LLM or VLM. Given the limitations of LLMs or VLMs, such as severe hallucination issues, how do the authors ensure the accuracy of the results obtained each time? For example, in Relevance Scoring, on one hand, Cap(.) is used to obtain captions for keyframes. How can we ensure that critical information is not lost? Additionally, using an LLM to judge relevance to the query, how can we ensure the accuracy of this relevance judgment? Furthermore, is it appropriate to filter and aggregate all video content based solely on the query? For instance, can a simple question like "Please describe the video content" be answered accurately?
4. This method involves a large number of hyperparameters.

**Questions:**

Please see the Weaknesses.

---

### Official Review · Reviewer_BoGB · 2024-11-03

**Soundness:** 3
**Presentation:** 3
**Contribution:** 2
**Rating:** 5
**Confidence:** 5

**Summary:**

This paper introduces VIDEOTREE, a training-free framework that enables large language models (LLMs) to perform efficient reasoning on long-form videos. VIDEOTREE builds a hierarchical, query-adaptive video representation by selectively extracting relevant keyframes and organizing them in a tree structure, where coarse-to-fine details are progressively refined based on query relevance. This approach reduces redundancy and informational overload, allowing the model to focus on pertinent video segments, thus improving reasoning accuracy and efficiency. Experiments show that VIDEOTREE outperforms existing training-free methods on long video benchmarks, achieving higher accuracy and faster inference times without additional video-specific training.

**Strengths:**

This paper, VIDEOTREE, introduces a tree-based structure that efficiently organizes video content, making it well-suited for handling long-form videos with complex temporal dependencies.

The tree-based model can be scaled to accommodate a wide variety of video lengths and complexities, making it versatile and adaptable across different domains.

**Weaknesses:**

The baseline methods evaluated in Video-MME are selected deliberately, the current version does not provide a comprehensive evaluation.

The performance of VIDEOTREE heavily relies on how well the video segments are defined. Inaccurate or suboptimal segmentation could impact the overall representation quality and LLM understanding.

The method assumes compatibility with existing LLMs for video processing, which may limit its effectiveness depending on the specific architecture and capacity of the LLMs being used.

**Questions:**

How does the hierarchical structure impact the LLM’s ability to reason across segments? Does the model see improvements primarily in short-term dependencies or in understanding the overall narrative?

Could VIDEOTREE be adapted for real-time video processing, or is it primarily suited to post-processing of pre-recorded content?

How well does VIDEOTREE generalize to different domains, such as instructional videos, movies, or surveillance footage? Are certain types of video content more compatible with the model's tree-based representation?

---

### Official Review · Reviewer_VKNm · 2024-11-03

**Soundness:** 3
**Presentation:** 4
**Contribution:** 2
**Rating:** 3
**Confidence:** 3

**Summary:**

This paper proposes VideoTree, a training-free framework that builds a query-adaptive and hierarchical video representation for LLM reasoning over long-form videos. Specifically, VideoTree first extracts query-relevant information from the input video through an iterative process, progressively refining the selection of keyframes based on their relevance to the query. Then VideoTree incorporates multi-granularity information into a tree-based representation, allowing the model to extract query-relevant details from long videos in a coarse-to-fine manner. Finally, VideoTree aggregates the hierarchical query-relevant information within the tree structure and feeds it into an LLM reasoning model to get the final answer.

**Strengths:**

(1) This paper presents an interesting idea by building a query-adaptive and hierarchical video representation to identify key frames.

(2) This paper is well-organized and the writing is clear.

**Weaknesses:**

(1) From a technical standpoint, the innovation is limited and can be categorized as incremental. Specifically, the query-adaptive visual cluster and the coarse-to-fine strategy for identifying keyframes have been explored in previous work.

(2) In the Introduction Section, the authors mention one of the motivations for VideoTree as the Inability to Capture the Coarse-to-Fine Video Structure. However, this motivation is unconvincing because capturing the coarse-to-fine video structure is merely a method for identifying key frames, rather than a true challenge faced in the field of long video understanding. It seems that the authors are somewhat justifying their proposed approach rather than addressing a broader, established challenge.

(3) In Section 3.1, the authors write that for a cluster $C_i$, they identify the keyframe $F_i$ closest to the centroid vector $\mathbf{c}_i$ and consider it as the keyframe of the $i$th cluster. However, this straightforward method of converting the image to a caption can result in a significant loss of information relevant to the query, leading to potential error propagation. Additionally, this image-level captioning operation overlooks substantial motion information, making it inadequate for addressing queries related to temporal dynamics.

(4) The experimental results on Video-MME are not inspiring and insufficient, and do not convincingly demonstrate the effectiveness of VideoTree. For instance, why not comparisons with similar methods (e.g., LLoVi, VideoAgent) ?  Additionally, it would be beneficial to see results that incorporate subtitles, as this could provide further insight into how VideoTree performs relative to existing approaches and under different input conditions.

**Questions:**

Please see weaknesses.

---

### Note · Authors · 2024-11-14

I have read and agree with the venue's withdrawal policy on behalf of myself and my co-authors.